# Intuitive Engineering as a Proxy Facilitates the Evolutionary Optimization of an Underactuated Robot Ball

Giacomo Spigler*, Simone Silenzi*, Bart Horstman, Heike Vallery

*Abstract*—Underactuation can enable low-cost, light-weight robotics. However, their design is challenging. While classical engineering intuition often leads to reasonable hardware and control choices that ensure basic functionality, the resulting performance is usually low. In contrast, purely data-driven co-evolution of hardware and software conventionally needs high computational effort to deliver meaningful results. We propose to leverage the advantages of both approaches by using classical intuitive controllers as proxies. As an example, we consider "Fizzy," an underactuated robotic ball that leverages a unique single-motor configuration in combination with dynamic imbalance for movement. In a first optimization, an intuitive Virtual Model Control (VMC) proxy serves to quickly evaluate various design parameters like motor mass and axle positioning for a Covariance Matrix Adaptation Evolution Strategy (CMA-ES). The optimized configurations then serve as a foundation for training more sophisticated deep reinforcement learning (DRL) controllers. Our methodology underscores the potential of integrating intuitive proxies with evolutionary algorithms to enhance the performance and efficiency of underactuated robotic systems, paving the way for more adaptable and cost-effective robotic designs.

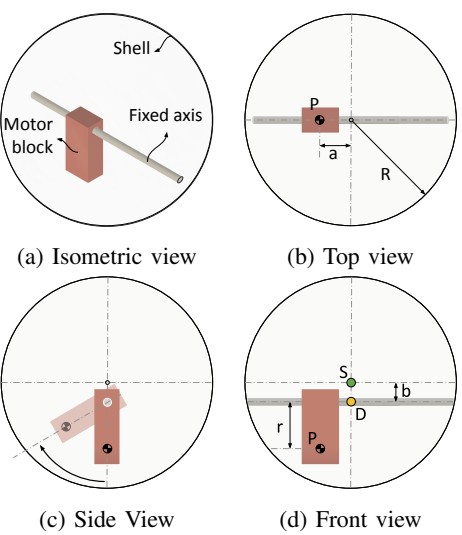

(a) Isometric view    (b) Top view

(c) Side View    (d) Front view

Fig. 1: Design of the Fizzy robot ball.

## I. INTRODUCTION

Underactuated robots -i.e. those with fewer actuators than degrees of freedom- are common in applications where lightweight design or cost are critical [1]. Fizzy is an example of such a lightweight, low-cost design: A robotic ball that rolls around using only one motor [2]. Its conceptual design is shown in Fig. 1. The ball's shell contains an off-centered axle around which a motorized block rotates. That way, the combined mass distribution of the ball changes dynamically. This construction detail ensures that the ball can move in the horizontal plane by using two effects: Displacement of the center of mass and inertial effects due to dynamic imbalance. That way, theoretically any point in the plane can be reached, similar to the solution by [3] for a unicycle. However, this economy in the design comes with higher control complexity.

Classical control methods for these systems are based on energy methods [4, 5, 6], on feedback linearization [7], or on Virtual Model Control (VMC) [8]. However, these methods are sensitive to parameters and noise and have limited regions of attraction.

Recent results for quadruped and biped robots based on Deep Reinforcement Learning (DRL) show that it is possible to learn robust strategies in a model-free way [9, 10]. We follow this direction and design two DRL controllers for Fizzy in a simulated MuJoCo [11] environment, and we compare them with a baseline VMC controller.

Since the performance of the robot strongly depends on its mechanical design parameters -i.e. the mass and inertia of the rotating motor block or the location of the block's axle- it is useful to co-optimize the mechanical design of the robot together with its controller [12, 13].

Such an approach has e.g. been previously proposed for legged robots in Chadwick et al. [14], where optimization methods are applied to the design of a robot's legs. Specifically, a metric based on motion trajectories together with forces and torques at the joints is optimized using Genetic Algorithms (GA) to find the best geometry for the robot legs. Another method based on GA is used for legged robots in Fadini et al. [15], where a two-stage optimization scheme allows for adjustment of the duration of the movement, the actuators, and the size of the robot. Other methods based on Covariance Matrix Adaptation Evolution Strategy (CMA-ES) [16] have been proposed by Hwangbo [17] for legged robots, and Maywald et al. [18] for an Acrobot, in order to increase the stability region of the controller and its closed-loop performance by tuning the parameters. Dinev et al. [19] instead optimize the morphology, payload distribution, and actuator parameters of a quadruped robot with a bilevel optimization approach, extracting the derivatives from the motion planner and using them for nonlinear optimization.

Nonetheless, co-optimizing a controller together with the

---

*These authors contributed equally to this work.

configuration of a robot body remains a challenging and computationally expensive problem, for two reasons. First, optimizing the body parameters of a robot over some metric typically requires the use of derivative-free optimization methods, like CMA-ES or GA, which are considerably less sample-efficient than gradient-based methods. Second, the evaluation of a controller based on DRL requires repeatedly training the controller from scratch for each robot configuration, which can take a long time (e.g., training a PPO agent on a simple Mujoco task can take around 5-30 minutes on a fast computer).

As an alternative approach, we suggest a three-stage process. We initially propose an intuitive design and create the controllers for this one. Then, rather than jointly optimizing the DRL controller and the robot configuration, we use a simpler -and faster- proxy controller in the body optimization stage, specifically VMC. Such a proxy is embodiment-independent and does not require adaptation to each possible configuration. We thus use VMC evaluate each robot configuration proposed by CMA-ES, optimizing for the average time it takes the robot to reach a set of randomized target locations. Only after optimizing the robot body, the DRL controllers are trained, using the final optimized configuration.

## II. METHODS

Our method has three parts. First, we manually design a basic controller using Virtual Model Control (VMC). Second, we train reinforcement learning controllers tailored for specific tasks. Third, we use evolutionary optimization to enhance the robot's configuration for optimal control performance. VMC serves as a proxy during optimization to save runtime. Afterwards, reinforcement learning is applied to the optimized robot to improve the performance versus the baseline controller.

### A. Modeling of Fizzy the Underactuated Robot Ball

Fig. 1 schematically shows the internal mechanism of Fizzy. An eccentric axle is attached to the shell and a motor block rotates around this axle. The motor block contains the battery, motor and electronics, as well as an inertial measurement unit (gyroscope, accelerometer, magnetometer).

The model consists of three objects of uniform density: The shell, the block, and the axle. Parameters are the ball's radius $R$, the axle eccentricity $b$, and distances $a$ and $r$ that describe the location of the center of mass $P$ of the block. The shell is simplified as a sphere of mass $m_s$, the block as a point mass of mass $m_b$, the axle as a slender rod of mass $m_a$.

### B. Baseline Controller: Virtual Model Control

The baseline Virtual Model Control (VMC) controller is designed in an intuitive way: It emulates a virtual spring that "pulls" the motor block towards a target $G$, effectively displacing the ball's overall center of mass and forcing the shell to follow. To mitigate singular configurations, where the motor axis is aligned with the target, a singularity index is calculated and used to modify the controller: The spring becomes stronger closer to the singularity and thereby helps the system "snap out" of that configuration.

Specifically, we define the singularity index $\delta$ as:

$$\delta = \left| \frac{\hat{\boldsymbol{b}}_1^T \, \boldsymbol{r}_{G/P}}{\lambda + \|\boldsymbol{r}_{G/P}\|} \right| \quad (1)$$

where $\boldsymbol{r}_{G/P}$ denotes the vector from point P to G, the unit vector $\hat{\boldsymbol{b}_1}$ points in the direction of the motor axle, and the normalization factor $0 < \lambda << 1$ prevents division by zero.

The VMC's virtual spring that "pulls" the motor block towards the goal is emulated by the motor torque $\tau$:

$$\tau = \text{sat}\left( \frac{K}{1-\delta} \, \hat{\boldsymbol{b}}_1^T \left( \boldsymbol{r}_{P/D} \times \boldsymbol{r}_{G/P} \right) \right). \quad (2)$$

The gain $K$ is the nominal stiffness of the spring, and $\text{sat}(\cdot)$ is the saturation function (here, $\text{sat}(x) = \text{clip}(x, -1, 1)$).

### C. Deep Reinforcement Learning

We formulate the control of the Fizzy robot via reinforcement learning by framing it as a Markov Decision Process defined by the tuple $(\mathcal{S}, \mathcal{A}, p, \mathcal{T}, r, \rho_\iota, \gamma)$ [20], with a finite time horizon $T$ that delimits different trials. Each trial starts by sampling a state from a distribution over initial states $s_0 \sim \rho_0(s)$ and proceeds at discrete steps running at 10Hz in a simulated Mujoco environment. At each step $t = 1, \ldots, T$ the agent picks an action $a_t \in \mathcal{A}$ according to its policy $\pi : \mathcal{S} \to \mathcal{A}$, which maps states $s_t \in \mathcal{S}$ to actions. The state of the MDP changes with each action according to the transition function $p : \mathcal{S} \times \mathcal{A} \to \mathcal{S}$, leading to new states $s_{t+1}$ together with scalar rewards that are sampled from the reward function $r : \mathcal{S} \times \mathcal{A} \to \mathbb{R}$. The objective of reinforcement learning is to find a policy that maximizes the accumulated rewards over episodes, discounted with factor $\gamma$:

$$\pi^\star(s) = \arg\max_\pi \mathbb{E}_\pi \left[ \sum_{t=1}^{T} \gamma^t r_t | a_t \sim \pi(s_t), s_0 \sim \rho_0(s) \right] \quad (3)$$

In this study, we use two reinforcement learning algorithms: Proximal Policy Optimization (PPO) [21], and Soft Actor Critic (SAC) [22]. We use standard implementations available in stable-baselines3 [23]. PPO, an on-policy method utilizing policy gradients, improves the stability of vanilla policy gradient by indirectly applying trust-region constraints indirectly to prevent policies to change too much at each training step. PPO optimizes the policy function directly using gradient descent while using a concurrently learned value function for variance reduction. Soft Actor Critic is an off-policy method that balances exploration and exploitation by optimizing a trade-off between reward collection and policy entropy.

### D. Goal-Reaching Environment

*1) Task:* We define a goal-reaching task in the following way: In each episode, the robot starts at the origin with a randomized configuration. That is, the orientation of the ball and motor are sampled uniformly at random, the initial velocities in the x and y directions follow a Normal distribution with a standard deviation of $0.5 \, \text{m/s}$, and the angular velocities

of both the ball and the motor are sampled from a Normal distribution with a standard deviation of $1\,\mathrm{rad/s}$.

A target location is sampled uniformly from a disc with a radius ranging from $0.5\,\mathrm{m}$ to $3\,\mathrm{m}$, centered at the origin. The robot's goal is to reach this target location as fast as possible by controlling the torque of its single motor.

A trial is considered successful if the robot reaches the target location (within a margin of $20\,\mathrm{cm}$) within 20 seconds.

*2) Observation Space:* The reinforcement learning agents receive a state representation composed of sensor readings (acceleration and angular velocities) from the IMU mounted on the motor block, together with the absolute rotation of the IMU as a quaternion. Additionally, the agent receives a goal vector indicating the relative Cartesian position of the target with respect to the center of the ball, expressed in the global coordinate frame.

*3) Reward Function:* To motivate the agent to reach the target location swiftly, we design a reward function as follows:

$$r(s_t, a_t, s_{t+1}) = 10\left(d(s_t) - d(s_{t+1})\right) - 0.5 \quad (4)$$

where $d(s_t)$ denotes the distance between the robot and the target at time $t$. Additionally, an extra reward of 200 is provided when the agent arrives at the target ($d(s_t) < 0.2\,\mathrm{m}$).

### E. Evolutionary Optimization with Proxy Controller

Our goal is to automatically optimize the configuration of the underactuated Fizzy robot to enhance its maneuverability and performance. To achieve this, we aim to adjust six key parameters, subsumed in vector $\boldsymbol{\theta} \in \mathbb{R}^n$, that influence the robot's dynamics. These parameters include the masses of the sphere $m_s$, axle $m_a$, and motor block $m_b$, as well as three distances: the lateral eccentricity of the motor block ($a$ in Fig. 1d), the distance $r$ between the motor block's center of mass $P$ and the motor axle, and the offset displacement of the axle from the sphere's center ($b$ in Fig. 1d).

We determine the effectiveness of different configurations by measuring the average time ($T_i$) needed for the VMC controller to reach a random target location across $n_{\mathrm{trials}} = 20$ trials. The VMC controller is thus used as a proxy to assess the quality of a configuration under the assumption that a configuration on which VMC performs well is also going to be easier to control for a DRL agent. Our objective is to find the parameters that minimize this average time

$$\boldsymbol{\theta}^\star = \arg\min_{\boldsymbol{\theta} \in \Theta} \frac{1}{n_{\mathrm{trials}}} \sum_{i=1}^{n_{\mathrm{trials}}} T_i. \quad (5)$$

Solving the problem above is difficult since it is not possible to analytically differentiate the objective with respect to the model parameters. We can however reframe the problem as derivative-free optimization, and then solve it using the Covariance Matrix Adaptation Evolution Strategy (CMA-ES) [16]. CMA-ES is an evolutionary algorithm that works by maintaining a multivariate normal distribution in $\mathbb{R}^n$, from which candidate solutions (parameter vectors $\theta$) can be sampled. The distribution is optimized via maximum likelihood to increase the probability of sampling higher-value solutions.

This involves alternating between improvement to the mean of the distribution and the covariance matrix.

The effectiveness of using a proxy controller becomes evident from expected computational effort for CMA-ES. For instance, with a population size of 200 and 50 CMA-ES iterations, we would need to evaluate $N = 200 \cdot 50 = 10,000$ robot configurations. If training a DRL agent for each evaluation took 20 minutes, the total training time would exceed 3300 hours (139 days). In contrast, evaluating the proxy controller takes less than 2 seconds per configuration, totaling around 5 hours, an approximate 600-fold increase in efficiency.

For complete details on the CMA-ES hyperparameters and the bounds $\Theta$ of the 6 parameters, please see Appendix A.

### F. Experimental Evaluation Setup

We evaluate the three controllers PPO, SAC, and VMC in a simulated Mujoco environment on a goal-reaching task.

The controllers undergo training on the same task in two phases: initially using the original Fizzy robot design, and then with a design optimized by CMA-ES. Detailed training procedures are described in Appendix A.

Controller performance is evaluated based on two main metrics: Success rate and median time to target. The success rate measures the proportion of trials in which the robot gets within $0.2\,\mathrm{m}$ of the target location within the allotted time. The median time to target reflects the typical episode duration -either the time taken to reach the target successfully, or 20 seconds if the target is not reached.

## III. RESULTS

### A. Performance of the Controllers

Results for the controllers trained on both the initial and the optimized robot designs are shown in Table I. The metrics shown are median values over 500 randomized trials. Square brackets denote the range of 90% of the values. A visualization of the success rate of the different controllers with respect to eight fixed targets is attached as Suppl. Fig. 5.

Our findings show that learning-based controllers (PPO and SAC) outperform the VMC controller in both metrics, achieving a higher success rate and lower times to reach the target locations, with SAC overall performing better than PPO. The result was consistent across the two robot embodiments.

We also find that the trajectories produced by PPO and SAC tend to approach the target directly, with fewer deviations and spread than the VMC controller. Fig. 2 illustrates this

|  | base model | | CMA-ES optimized model | |
|---|---|---|---|---|
|  | success rate | time to target / s | success rate | time to target / s |
| ppo | 92% | 9.6 [2.45, 20] | **100%** | 2.6 [1.1, 4.65] |
| sac | **93%** | **7.3** [1.35, 20] | **100%** | **2.3** [1, 4.1] |
| vmc | 46% | 20 [1.3, 20] | 99.4% | 3.2 [1.1, 7.5] |

TABLE I: Performance of the three controllers PPO, SAC (reinforcement learning), and VMC (manually designed) on goal-reaching tasks for the Fizzy robot ball.

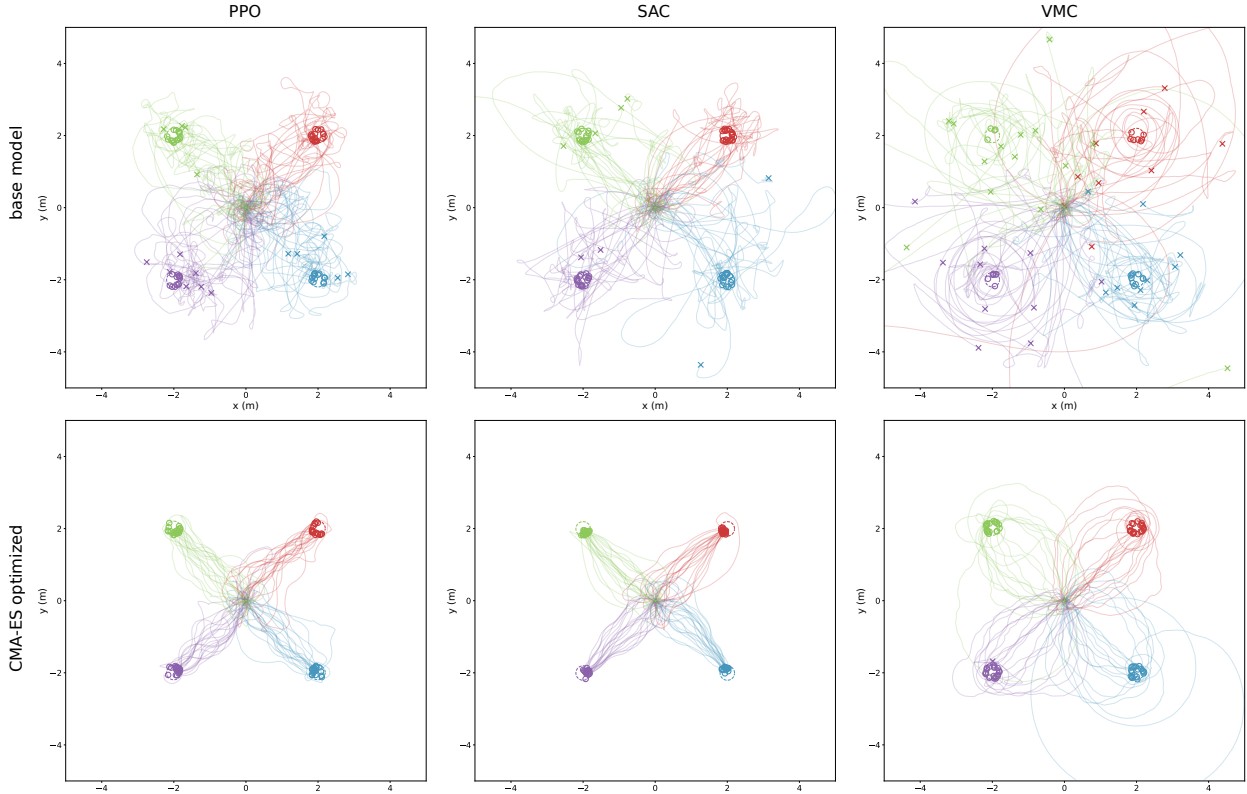

Fig. 2: Trajectories of the Fizzy robot controlled with the PPO, SAC, and VMC controllers. The task involves guiding the robot to 4 goal locations, marked with different colors. For each goal, we collect 20 trajectories by resetting the robot to the origin and randomizing its initial state. The *top row* shows trajectories from controllers based on the initial robot body, while the *bottom row* shows those based on the robot body optimized through CMA-ES. Small circles indicate the robot's position if it successfully reaches a goal, whereas crosses mark its last position if time ran out.

difference, depicting trajectories to four designated targets, each tested 20 times under randomized initial conditions.

### B. Evolutionary Optimization of the Robot Body

Optimization with CMA-ES approached convergence within the first 20 iterations of optimization, with performance improving monotonically (see Suppl. Fig. 3). Both original and optimized parameters are provided in Table II.

We then train new PPO and SAC agents on the optimal robot found by CMA-ES, and evaluate the new controllers together with VMC. The results (Table I) show that all controllers achieve considerably better performance on the optimized robot than in the original one, with times to target $\sim$ 3-6 times lower, and with a success rate close to $100\%$.

### IV. DISCUSSION AND CONCLUSION

We found that Deep Reinforcement Learning considerably improved the control performance of the Fizzy robot compared to the intuitively designed Virtual Model Control.

Using VMC as a proxy controller while optimizing the configuration of the robot, rather than optimizing directly for the performance of DRL controllers, proved crucial. This approach drastically cut computational costs and time, showing

|  | $m_s$ / g | $m_a$ / g | $m_b$ / g | $a$ / cm | $r$ / cm | $b$ / cm |
|---|---|---|---|---|---|---|
| original | 150 | 27 | 150 | 1.36 | 3.16 | 2.48 |
| optimized | 56.9 | 149.1 | 276.5 | 0.13 | 4.87 | 0.15 |

TABLE II: Original and optimized parameters: $m_s$ shell mass, $m_a$ axle mass, $m_b$ motor block mass, $a$ and $b$ as shown in Fig. 1d, and $r$ distance between the CoM of the motor and the axle.

the effectiveness of simplified models in the early stages of optimization. The fact that PPO and SAC still outperformed VMC, despite the robot body being optimized explicitly for VMC, highlights the robustness and flexibility of RL.

This simulation was a proof of concept, and future work will e.g. include more realistic modeling of the mechanical system and its sensor processing. Nonetheless, it shows the potential of machine learning techniques for controlling underactuated systems, highlighting the intricate interplay between mechanical design and control strategies.

The benefits of integrating evolutionary algorithms with Deep Reinforcement Learning may also transfer to other robots, especially underactuated ones.

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

## A. Supplementary Methods

*1) CMA-ES Optimization:* We use the CMA-ES implementation provided in the *cmaes* Python package with parameters $\sigma = 0.1$ and population size $n = 200$, and perform 50 iterations of optimization. We enforce constraints on the individual parameters $m_s$ (mass of the shell), $m_a$ (mass of the axle), $m_b$ (mass of the motor block), $a$ (lateral displacement of the motor block), $r$ (distance between the center of mass of the motor block and its axis of rotation), and $b$ (offset displacement of the axis of rotation from the center of the sphere). The upper and lower bounds for each parameter are shown in Table III.

To improve on the physical realism of the configurations, we further enforce a penalty for configurations where the center of mass of the motor block can reach closer than $3\,\mathrm{cm}$ from the shell, to prevent the motor block from hitting the shell. The penalty is enforced by giving a constant penalty of $20\,\mathrm{s}$ to the configuration, corresponding to the worst possible score.

*2) DRL Training and Hyperparameters:* DRL agents were trained with the following hyperparameters. Both SAC and PPO used distinct actor and critic networks with the same architecture. First, features are learned through MLP embeddings for the state vector (two hidden layers of size 256 and 128) and the task information (relative distance of the target with respect to the agent; two hidden layers of size 32 and 32). The embeddings are then concatenated and input into two separate MLPs -one for the policy and one for the value (PPO) or Q function (SAC)- each with 128 units. Hyperbolic tangent was used as an activation function for all hidden layers.

The training was performed over 5M timesteps using 16 parallel environments. Observations and rewards are normalized using running statistics. The hyperparameters used to train the SAC agents are: $\gamma = 0.995$, and 2 gradient updates are performed for each environment timestep. The learning rate was set to $\eta = 0.0003$, the size of the experience replay buffer was 1000000, updates were calculated with a batch size of 256, and the target network tracked the main network via soft updates with $\tau = 0.005$. The hyperparameters of PPO-clip are: $\gamma = 0.995$, batch size 256, rollouts of 256 steps, and clip $\epsilon = 0.2$. The training was performed over 3 epochs for each environment timestep, using minibatches of size 256 and learning rate $\eta = 0.0003$. Generalized Advantage Estimation was used with $\lambda = 0.95$.

## B. Supplementary Results

Fig. 3 reports the training curve of CMA-ES, showing a monotonic decrease in the mean time to target the configurations samples from the population during training.

Performance of the DRL agents is shown in Fig. 4.

Fig. 5 is produced in a manner similar to Fig. 2, but with 8 target locations, a larger number of trials per goal, and without drawing the individual trajectories, to remove clutter. The figure shows the success rate and the precision of the three controllers in the two embodiments. We also note that, while all controllers achieve a 100% success rate on the optimized

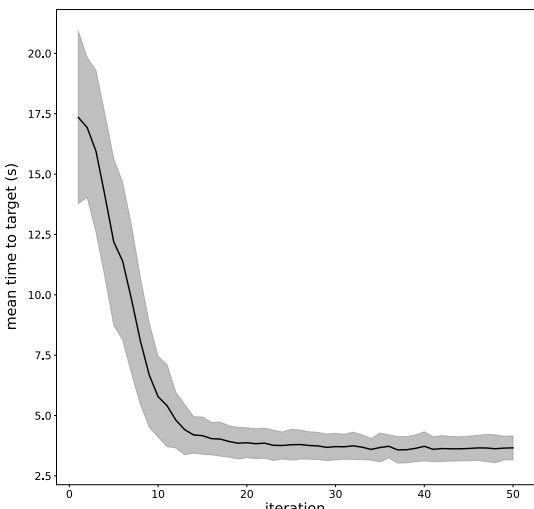

Fig. 3: The mean time to reach the target is plotted for each optimization iteration, averaged over candidate solutions sampled from the CMA-ES distribution at each iteration. The shaded area denotes the standard deviation.

| | $m_s$ / g | $m_a$ / g | $m_b$ / g | $a$ / cm | $r$ / cm | $b$ / cm |
|---|---|---|---|---|---|---|
| lower | 50 | 10 | 50 | 0 | 1 | 0 |
| upper | 500 | 200 | 300 | 4 | 5 | 5 |

TABLE III: Bounds $\Theta$ used as constraints for CMA-ES optimization.

robot, SAC manages to reach the target in a more straight way, as evidenced by the final positions of the robot being closer to the origin.

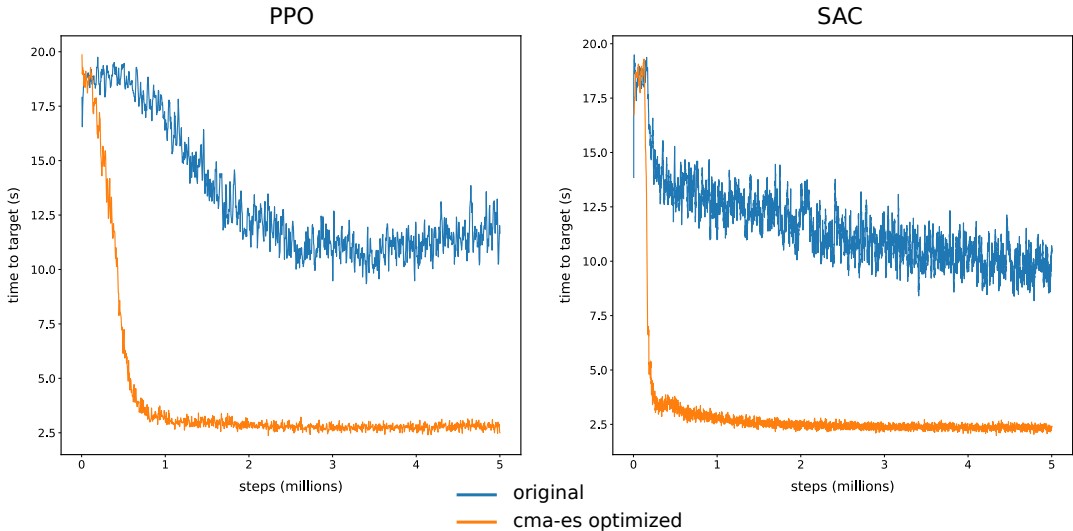

Fig. 4: Training curves for the two reinforcement learning agents (PPO and SAC) trained in both the original (blue) and CMA-ES optimized (orange) robot embodiments.

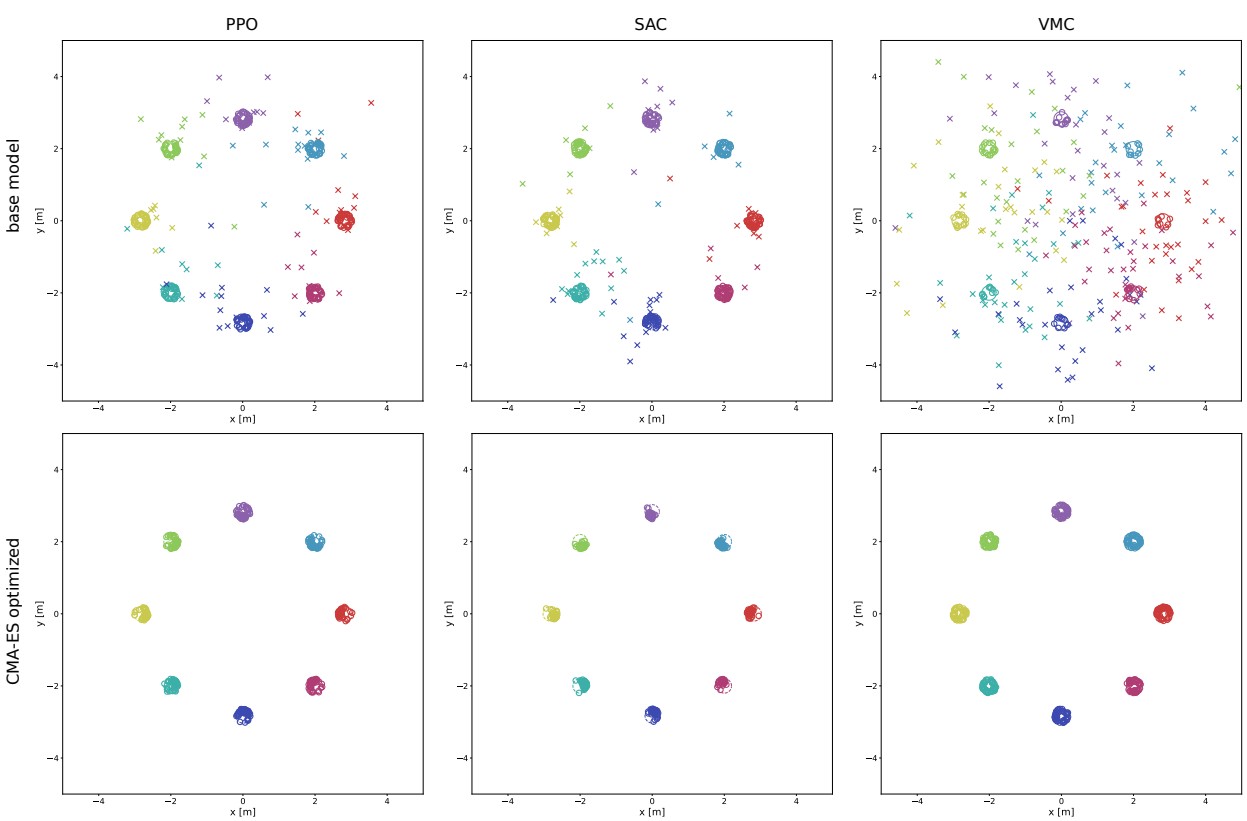

Fig. 5: Final position of the Fizzy robot ball controlled with the three controllers PPO, SAC, and VMC, on tasks where the robot is required to reach 8 goal locations. For each target location (indicated by a different color), we collect 50 trajectories by resetting the robot to the origin and randomizing its initial velocity, angular velocities, and pose. The *top row* shows trials from controllers trained and evaluated on the initial robot body, while the *bottom row* shows the controllers trained and evaluated on the robot body optimized through CMA-ES. Small circles are used to denote the final position of the robot if it manages to reach its designated goal, while crosses denote the last position of the robot when the 20 seconds timeout is reached.