# OpenReview forum: "Intuitive Engineering as a Proxy Facilitates the Evolutionary Optimization of an Underactuated Robot Ball"
_roboticsfoundation.org/RSS/2024/Workshop/EARL — EARL 2024 Poster_

### Official Review · Reviewer_jA7y · 2024-06-22
**A convincing proposal to accelerate the co-design of hardware and control**

**Rating:** 7
**Confidence:** 4

**Review:**

The co-design of robot morphology and controllers is an important problem in under-actuated robotics, but this can be very time-consuming if the control policy needs to be nontrivial (and thus learnt from data, using e.g. Deep RL).

The authors propose a very well-explained treatment of this problem for the case of a single-motor-driven spherical robot ball --- called “Fizzy” --- that must navigate to desired 2-D floor coordinates.This robot has only 6 morphological parameters to be evolutionarily optimized through CMA-ES, with the objective of making the controller more successful.

Doing this optimization in tandem with RL training is expected to be extremely costly due to the long training times of the policies (which must be re-trained after each morphology update), and therefore suggest an alternative, quite reasonable idea: they use CMA-ES to optimize the performance of a different, hand-engineered heuristc controller (VMC) that is not trained, allowing very fast evaluations; and then train two different RL agents (PPO and SAC) separately on the final morphology, in the hopes that a morphology that lends itself to better task success under a heuristic controller is likely to also be well-suited for RL.

The results indeed show that the RL agents (as well as VMC) perform better when trained with the optimized morphology, than with the original robot design, despite the optimization having only targeted VMC.

While this is a convincing and valuable insight, the experiments are still limited to one toy setting, and I think the paper can be made stronger by including additional experiments, with diverse robots and model-based baseline controllers like Mujoco-MPC.

---

### Decision · Program_Chairs · 2024-06-24

Accept (Poster)